# Vitamin D Metabolic Pathway Components in Orthopedic Patientes—Systematic Review

**DOI:** 10.3390/ijms232415556

**Published:** 2022-12-08

**Authors:** Janusz Płomiński, Roman Grzybowski, Ewa Fiedorowicz, Edyta Sienkiewicz-Szłapka, Dominika Rozmus, Angelika Król-Grzymała, Beata Jarmołowska, Natalia Kordulewska, Anna Cieślińska

**Affiliations:** 1Clinical Department of Trauma—Orthopedic Surgery and Spine Surgery of the Provincial Specialist Hospital in Olsztyn, 10-561 Olsztyn, Poland; 2Department and Clinic of Orthopedics and Traumatology, Collegium Medicum, University of Warmia and Mazury, 10-719 Olsztyn, Poland; 3Faculty of Biology and Biotechnology, University of Warmia and Mazury, 10-719 Olsztyn, Poland

**Keywords:** vitamin D, cholesterol, VDBP, CYP24A1, CYP27B1, VDR, orthopedics, ACL, RC, arthroplasty

## Abstract

Vitamin D takes part in the functioning of many processes that ensure the homeostasis of the body. In orthopedics, it is indicated as an inseparable element ensuring proper bone growth and functioning, and its deficiencies are indicated in various diseases, mainly in the proper structure and function of the skeleton. In this review, we focus on the most important components of the vitamin D metabolic pathway, in correlation with selected orthopedic conditions. Records were obtained from the PubMed database in a timeline of 2010–2022. The keywords were as follows: vitamin D/cholesterol/vitamin D binding protein/ VDBP/Cytochrome/CYP24A1/CYP 27B1/Vitamin D receptor/VDR/ + diseases (ACL reconstruction, rotator cuff, arthroplasty knee/hip/shoulder). The recent original studies were analyzed, discussed, and the most important data were shown. The vast majority of articles concern the metabolite of vitamin D (25(OH)D), which is measured as a standard in diagnostic laboratories. Even though there is a lot of valuable information in the literature, we believe that the other elements of the vitamin D pathway also deserve attention and suggest their research in correlation with orthopedic disorders to supplement the missing knowledge on this topic.

## 1. Introduction

The aim of this paper is to present the recent data gathered from 2010 to 2022. We discussed articles that seem to include connections between vitamin D pathway metabolites (also proteins) and disease. Despite finding valuable information in the literature, we found that there is not much data that connects all of our subjects of interest: vitamin D/cholesterol/vitamin D binding protein/VDBP/Cytochrome/CYP24A1/CYP 27B1/Vitamin D receptor/VDR/ + disease (ACL reconstruction, rotator cuff, arthroplasty knee/hip/shoulder) rarely covering the disease with enlarged findings in the vitamin D metabolic pathway. How is the vitamin D metabolic pathway connected to orthopedic diseases? How does vitamin D and its metabolites affect the possibility of injury? How does cholesterol affect vitamin D levels? How does the concentration of the most frequently studied metabolite, 25OHD, affect the concentration of subsequent metabolites, engaged proteins, receptors, and their subsequent effects on the organism? What is/are the most important factor(s) of the vitamin D metabolic pathway associated with a higher risk of diseases and extended treatment process?

The term vitamin D (VD) applies to the fat-soluble 9,10-secosteroids whose active metabolites are capable of binding the intracellular vitamin D receptor (VDR). These include vitamin D3 (cholecalciferol) produced by animals and vitamin D2 (ergocalciferol) produced by plants and fungi. Vitamers differ slightly in the structure of the side chain at the cholesterol moiety. This generates small differences in their pharmacokinetics, but in general, they show similar biological activity after activation [1]. Unlike most other vitamins, vitamin D3 can be produced in the human body from 7-dehydrocholesterol. This process takes place in the keratinocytes of the epidermis reproductive layer, but only after sufficiently long exposure to sunlight or UVB radiation. Apart from the availability of the substrate itself (7-DHC), the second important factor limiting the synthesis of vitamin D3 is the photoisomerization of 7-DHC. This reaction is not subjected to internal regulation and largely depends on external factors. Although it is believed that under favorable conditions the endogenous synthesis of cholecalciferol may cover even 80–100% of the daily requirement for this vitamin, in practice the available data indicates that VD deficiency is a global public health problem [2,3]. The second, but much less efficient source of vitamin D, is diet. The mechanisms of vitamin D intestinal absorption remain still poorly understood. Passive diffusion is observed with pharmacological doses of VD, but recent in vivo experiments and in vitro studies demonstrated that it could be absorbed also by mechanisms involving membrane carriers, in a way similar to the diet-derived cholesterol [4]. Absorbed VD is incorporated into chylomicrons that reach the systemic circulation via the lymphatics. From the circulation it is taken up by adipose tissue and skeletal muscle (accounted for the rapid postprandial clearance of vitamin D from plasma) or delivered with chylomicron remnants to the liver. Mechanisms of VD accumulation or later mobilization are not clear, but it is suggested that adipose tissue represents rather “non-specific” stores where VD is sequestered due to its hydrophobic nature. These deposits are not readily available but could prevent the body from reaching critically low levels of VD concentration in a time of vitamin deprivation [5,6]. Both vitamin D3 produced in the skin and vitamin D2 and D3 absorbed in the intestine are biologically inactive and undergo the same metabolism in the body. The first step of activation is conducted in the liver by a group of 25-hydroxylases (CYP2R1, CYP27A1, CYP3A4), with CYP2R1 being the main enzyme responsible for maintaining the physiological level of 25-hydroxyvitamin D [7,8]. The resulting 25(OH)D (D3—cholecalcidiol or D2—ergocalcidiol) is transported to the kidneys where it undergoes a second hydroxylation (CYP27B1) to the final active form of vitamin 1,25(OH)2D (D3—cholecalcitriol or D2—ergocalcitriol) capable of activating the VDR receptor. This route is considered to be the main pathway of vitamin D activation in the human body, but the activity of 25-hydroxylases, and especially 25-hydroxyvitamin D 1α-hydroxylase (CYP27B1), is also commonly found in peripheral tissues. The peripheral production of 1,25(OH)2D provides locally 100–1000 times higher concentrations of the active metabolite than its plasma concentration, but under physiological conditions, it does not affect its level in the blood [9,10]. The active metabolite 1,25(OH)2D undergoes 24-hydroxylation carried out by broadly distributed CYP24A1, which initiates the process of its inactivation and elimination in the form of biologically inactive calcitroic acid. Since CYP24A1 is also able to 24-hydroxylate 25(OH)D, directing it to the catabolic pathway before full activation, this enzyme is currently indicated as one of the main factors preventing the toxic effects of vitamin D excess [11].

The liver metabolite 25(OH)D, due to its greater stability (half-life 2–3 weeks) and relatively high concentration values (ng/mL) compared to the renal metabolite 1,25(OH)2D (half-life 4–6 h, concentration in pg/mL), is currently considered to be the best indicator of the body’s supply of vitamin D [12]. Vitamin D and its metabolites are transported in the blood by specific vitamin-D-binding protein (VDBD) (~85%) and serum albumin (~15%). Only little amounts of 25(OH)D (~0.03%) and 1,25(OH)2D (~0.4%) are found in the plasma in a completely free form. Due to the low affinity of VD metabolites to albumin (Ka for 25(OH)D = 6 × 10^5^ M^−1^ and for 1,25(OH)2D = 5.4 × 10^4^ M^−1^), the free and albumin fraction are considered as “easy bioavailable”, while the role of VDBP is to be a reservoir of the VD metabolites, especially the liver metabolite (Ka for 25(OH)D = 5 × 10^8^ M^−1^, and for 1,25(OH)2D and VD = 4 × 10^7^ M^−1^) [13].

The active metabolite of vitamin D, 1,25-dihydroxycholecalciferol, works through a specific vitamin D receptor (VDR) [14]. Due to the fact that vitamin D acts through VDR receptors, their impairment or reduced functionality may have a crucial impact on the balance of the vitamin D concentration, and the final metabolite activity throughout the body. VDR is expressed in nerve cells, glial cells, and cells of the immune system, such as monocytes, macrophages, or activated T and B lymphocytes, as well as in cancer cells (colon cancer) and liver stellate cells. The presence of these receptors allows for the regulation of gene expression involved in organ development, cell cycle control, calcium and phosphate homeostasis in bone metabolism, and xenobiotic detoxification. Vitamin D receptor (VDR), enzymes, and metabolites have different expression in various types of immune cells such as lymphocytes, monocytes, macrophages, and dendric cells [15,16]. Vitamin D receptors have been found in many tissues, where they perform several different functions related to the regulation of calcium–phosphate metabolism and maintaining the proper structure and function of the skeleton. Figure 1 presents the metabolic pathway of vitamin D with the most important components.

## 2. Methods

Literature search strategy: here we present information on the vitamin D metabolic pathway in human orthopedic diseases, and the principal references were taken from Internet databases as PubMed and Google Scholar published from January 2010 till June 2022. The keywords were as follow: vitamin D/cholesterol/vitamin D binding protein/VDBP/Cytochrome/CYP24A1/CYP 27B1/Vitamin D receptor/VDR/ + diseases (ACL reconstruction, rotator cuff, arthroplasty knee/hip/shoulder).

We included original studies that described the components of the vitamin D metabolic pathway in common orthopedic diseases. We reviewed all publications in English and those having English abstracts. Figure 2 shows the scheme of literature searching methodology.

We present the collected information in the form of subsections with the names of found orthopedic injuries/diseases and then with a description, data collection, and table summary.

All the reviewers worked independently on every disease and metabolite/protein, gathering data. All articles were discussed whether to accept or reject it. Articles removed were duplicates, no vitamin D metabolites concentration or disease not specified/connected to our topic of research. We decided to present knowledge from 2010 as we wanted to focus on the newest data. We have omitted the information about the lack of some metabolites in the described disease. Articles not found about “metabolite + disease” have been removed under the subsection. There are also different results in different articles, but we did not decide to remove the contradictions, because our task was not to narrate the validity of the results, but to draw attention to the differences and indicate the need to unify this knowledge by deepening the research and describing the methods used more widely, as well as standardizing research in the context of vitamin D seasonality.

We did not exclude any study because the source of financing/bias.

## 3. Results and Discussion

### 3.1. Total Knee Arthroplasty (TKA)

There is increasing evidence regarding the effects of vitamin D on the proper functioning of skeletal, muscle, and physical function. Vitamin D is also considered as a beneficial nutrient for bone metabolism that can influence arthroplasty survival. Many researchers are investigating whether patients with low serum vitamin D levels may have functional abnormalities after total knee arthroplasty (TKA).

#### 3.1.1. Vitamin D

Shin et al. (2017) [17] conducted a prospective cohort study involving 92 patients who were divided into two groups according to the vitamin D levels assessed at the preoperative visit: (1) the group with vitamin D deficiency (levels < 12 ng/mL); (2) the group without vitamin D deficiency (level ≥ 12 ng/mL). Assessment of postoperative function was estimated using the following parameters: American Knee Society score (KSS), alternative step test (AST), six-meter walk test (SMT), sit-to-stand test (STS), and timed up-and-go test (TUGT). Patients with normal preoperative vitamin D levels showed better results in physical activity; therefore, it seems that vitamin D deficiency adversely affects early postoperative functional results after TKA [17]. On the other hand, Maniar et al. (2016) [18] proved total knee arthroplasty (TKA) should not be delayed among patients with vitamin D deficiency. The authors described that the use of appropriate postoperative supplementation was able to improve recovery in patients. It was an effect of research conducted among 120 patients. Vitamin D deficiency was detected in 64 participants before the TKA (25-hydroxyvitamin D < 30 ng/mL), but after 3 months of supplementation, all functional results were similar to those in patients without preoperative deficiencies [18]. The need for vitamin D supplementation in postoperative patients has also been emphasized by Mouli et al. (2022) [19], who carried out a retrospective analysis of 174 patients with vitamin D (25(OH)D) deficiency at a level <30 ng/mL. Participants were administered preparations according to the following schedule: daily supplementation of vitamin D3 (from 1000 to 6000 IU) or a loading dose of 50,000 IU weekly for 4 weeks, and then 2000 IU/d. The authors concluded that the second approach is more effective in correcting vitamin D deficiency; thus, supplementation should be used in patients prior to TKA [19].

Kong et al. (2021) [20] investigated the relationship between calcium and vitamin D use and the revision rate after primary total knee replacement surgery. The authors performed a population cohort study using the Korean National Health Insurance database that included all patients who underwent primary total knee replacement between 2009 and 2018. A total of 142,147 patients were included in the analysis and only about 20% of the respondents (n = 28,403) took preparations containing vitamin D and calcium. It has been proven that implant survival was significantly better in people taking a combination of calcium and vitamin D (at a dose of 800 IU or more) for more than 1 year compared to those who had never used it (*p* < 0.001) [20]. Song et al. (2021) [21] undertook to investigate the effect of vitamin D levels on early clinical function outcomes and potential risk factors for moderate to severe pain in postmenopausal women after TKA. The study included 226 women who were included in the group: the group with normal serum vitamin D concentration (level ≥ 30 ng/mL) and the group with vitamin D deficiency (level < 30 ng/mL). Although there was no significant difference in preoperative assessment of clinical function between the two groups, the authors concluded that vitamin D deficiency may adversely affect early functional outcomes in postmenopausal women after TKA and may be considered as a risk factor for moderate to severe postoperative knee pain (in correlation with smoking and high mass index) [21].

#### 3.1.2. Cholesterol

Cholesterol is a lipid from the group of steroids, also classified as an alcohol. In the biochemical approach, cholesterol is a precursor of bile acids, sex hormones, and vitamin D. It is known that vitamin D is important in the proper functioning of the skeletal system; therefore, the role of the precursor, which is cholesterol, should be intensively studied and correlated with orthopedic diseases.

Grey et al. (2005) [22] analyzed results from 20 patients after an elective knee arthroplasty. Blood was obtained postoperatively after 12, 24, 48, 72, and 168 h, and used for concentration determination of C-reactive protein, albumin, cholesterol, triglycerides, and malondialdehyde, as well as the lipid-soluble antioxidant vitamins. There was a significant increase in the concentration of circulating C-reactive protein (peak at 48 h, *p* < 0.001) and a significant decrease in the concentration of albumin, cholesterol, and triglycerides at the same time point (*p* < 0.001) during the whole study period. The authors indicated that in seemingly healthy individuals subjected to acute inflammation, circulating fat-soluble vitamin antioxidants were temporarily reduced [22]. However, after lipid adjustment, the concentrations were similar to the baseline values. Similar studies were conducted by Zhang et al. (2021) [23] who analyzed both high-density lipoprotein cholesterol and apolipoprotein A1 in the synovial fluid as potential prognostic factors for the severity of primary osteoarthritis of the knee. The study included 184 patients with osteoarthritis who received arthroscopic debridement or total knee arthroplasty, and 180 healthy controls. The biological material was the serum and synovial fluid in which it was determined: total triglycerides (TG), total cholesterol (TC), high-density lipoprotein cholesterol (HDL-C), low-density lipoprotein cholesterol (LDL-C), apolipoprotein A1 (ApoA1), and apolipoprotein B (ApoB). Any significant differences were not determined in serum TG and ApoB concentrations between both groups, while primary knee osteoarthritis patients had higher TC and LDL-C levels, and lower HDL-C and ApoA1 levels (*p* < 0.05). HDL-C and ApoA1 levels in synovial fluid were negatively correlated with cartilage damage, radiographic severity, and symptomatic severity of primary knee OA; therefore, the authors suggest that these factors be considered potential biological markers [23].

#### 3.1.3. Vitamin D Binding Protein (VDBP)

It is widely known that low vitamin D levels are characteristic of patients undergoing total hip arthroplasty (THA) and total knee arthroplasty (TKA), but it is still unclear whether the outcome of such surgery is related to vitamin D levels [24]. Probably only a small fraction of 25(OH)D remains free or unbound in plasma (0.02–0.05%), while 80–90% of vitamin D is circulating with vitamin D binding protein (VDBP) and 10–20% bound with albumin. The plasma half-life of VDBP is measured in days, and its concentration should be considered to decrease with cell damage and tissue loss [24].

There are indications that vitamin D status at surgery appears to be a better predictor of long-term surgical outcomes compared to postoperative metabolite level, as the latter may be influenced by perioperative conditions. The authors suggest that surgical trauma during THA or TKA and an increase in C-reactive protein (CRP) determine a change in metabolic status, including blood vitamin D levels [24]. Other scientists analyzed whether plasma 25(OH)D concentrations changed during the evolution of the systemic inflammatory response. Venous blood samples were collected preoperatively and postoperatively (starting 6–12 h postoperatively and every morning for 5 days) in 33 patients (range of age: 52–81 years old) who underwent primary knee arthroplasty. Then, 25(OH)D, vitamin D binding protein (VDBP), parathyroid hormone (PTH), calcium, C-reactive protein (CRP), and albumin were determined. It was found that preoperatively most of the patients were 25(OH)D deficient (<50 nmol/L) and had secondary hyperparathyroidism (PTH > 5 pmol/L), and simultaneously CRP and 25(OH)D had not returned to preoperative concentrations by 5 days postoperatively (*p* < 0.001). Moreover, VDBP and albumin were shown to be significantly reduced by 15% at day 1 (*p* < 0.001). Although VDBP levels returned to preoperative levels during the 5-day postoperative study period, this mechanism was not observed for albumin. The molar ratio and calculated free 25(OH)D concentration decreased significantly by about 40% within 24 h and did not return to preoperative values (*p* < 0.001). The authors therefore suggest that a molar ratio of vitamin D to VDBP should be considered to provide a useful indicator of biological activity [24]. The biological mechanism of this process can be explained as follows: it is known that stress (including surgical stress) reduces the concentration of most proteins in the serum and requires the rapid regulation of protein-dependent hormones. Vitamin D activation by 1α-hydroxylation of the kidneys is dependent on protein binding since 1,25-dihydroxyvitamin D (1,25(OH)2D3) is formed after the reabsorption of 25-hydroxyvitamin D (25OHD) bound to vitamin D binding protein (DBP) mediated by megalin. Postoperative changes in serum DBP and albumin concentration may therefore impair 1,25(OH)2D3 production [25]. The aim of the studies performed by Blomberg Jensen et al. (2018) was to determine sex-specific changes in the concentration of vitamin D metabolites and sex steroids in the serum 2, 6, 24, and 48 h and 3 weeks after the surgery. A total of 14 women and 11 men (aged from 45 to 77 years) without severe comorbidities undergoing unilateral total knee arthroplasty (TKA) were selected for the study, and then serum analysis was performed for total and free serum concentrations of 25OHD, 1,25(OH)2D3, 24,25-dihydroxyvitamin-D, DBP, albumin, sex hormone binding globulin (SHBG), calcium, and parathyroid hormone (PTH). As a result of the research, the authors showed that, the concentration of serum albumin and SHBG decreased postoperatively. Unexpectedly, the concentrations of DBP and 25OHD remained unchanged, but 1,25(OH)2D3 decreased after surgery. 1,25(OH)2D3 was lower (about 24%) 3 weeks postoperatively compared to preoperative level, while 24,25-dihydroxyvitamin-D was unchanged in postmenopausal women. The calculated conversion ratio of 25OHD to 1,25(OH)2D3 was strongly related to the preoperative effects of serum 25-OHD and PTH, while serum calcium was the most predictive after surgery. The authors propose that a critical and rate-limiting step in vitamin D activation is 1α-hydroxylation of the kidneys, which is tightly regulated by PTH, fibroblast growth factor 23 (FGF23), sex steroids, interferons, and other factors. It is worth noting that all circulating vitamin D metabolites are transported by vitamin D binding protein (DBP) or serum albumin, and the freely available vitamin D metabolite levels are significantly lower than other steroid hormones due to the strong binding to DBP [25].

#### 3.1.4. Cytochrome-P450-Mediated Metabolism of Vitamin D

An important role in the metabolism of vitamin D is played by sterol hydroxylases containing cytochrome P450 (CYP), whose function is to generate and degrade the active form of vitamin D3, which serves as a ligand for the expression of transcription genes through the vitamin D receptor (VDR) [9]. Incorrect activity of hydroxylase and metabolic blocks in the vitamin D pathway may result in the inability to use it in biochemical processes. Therefore, the implementation of biological functions related to vitamin D in our organism depends not only on its concentration, but also on factors such as VDR, VDBP, and hydroxylases such as CYP27B1, CYP24A1, and CYP2R1. It is known that the direct effect of vitamin D may be correlated with the specific bone cell types: osteoblasts, osteocytes, and osteoclasts expressing genes for vitamin D metabolizing enzymes, 25-hydroxyvitamin D 1∝-hydroxylase (CYP27B1), the enzyme that catalyzes the conversion of 25(OH)D to 1,25(OH)2D, and 25-hydroxyvitamin D 24-hydroxylase (CYP24A1), the enzyme that catalyzes the degradation of 1,25(OH)2D as well as vitamin D receptor (VDR) [26,27]. Sharma et al. (2019) described the role of these enzymes and serum 25-hydroxyvitamin D levels in association with improved bone formation and micro-structural measures in elderly hip fracture patients. Serum 25(OH)D, 1.25(OH)2D, and parathyroid hormone (PTH) levels were tested by immuno-enzymatic tests, while bone mRNA levels for vitamin D metabolizing enzymes CYP27B1 and CYP24A1 were measured by qRT-PCR. The authors demonstrated that the levels of CYP27B1, CYP24A1, and VDR mRNA are of direct importance in the regulation of vitamin D involvement in bone remodeling. Additionally, they confirmed that the corresponding serum 25(OH)D levels together with the bone mRNA level, which is higher for CYP27B1 and lower for CYP24A1, are associated with the formation of stronger trabecular bone structure [27]. The association of CYP27B1 and CYP24A1 activity with the concentration of 1.25(OH)2D3 was also emphasized by Blomberg Jensen et al. (2018). They performed analyses on 25 patients undergoing total knee arthroplasty (TKA). It has been noted that persistently high CYP24A1 activity in women concomitantly with lower CYP27B1 levels means that a greater proportion of the substrate pool is inactivated due to increased CYP24A1 activity, which alone or in combination with decreased CYP27B1 activity may result in the low 1.25(OH)2D3 [25].

#### 3.1.5. Vitamin D Receptor (VDR)

Vitamin D deficiency is common and affects millions of people worldwide. Therefore, it is normal to find hypovitaminosis in a variety of orthopedic populations, including trauma and arthroplasty. However, we do not know exactly whether this phenomenon only reflects the overall prevalence of vitamin D deficiency, or whether it affects the outcome of certain pathologies in specific populations at risk [28]. It is known that the biological action of vitamin D is closely related to the attachment to the vitamin D receptor (VDR). Upon binding of the VDR, it forms a heterodimer with the retinoid X receptor (RXR), which then facilitates the translocation of the VDR from the cytoplasm to the nucleus. The complex then binds to vitamin D responsive elements in the regulatory region of the target genes. We now know that almost all tissues and cells in the body have VDRs and that several of them have the enzymatic ability to convert the primary circulating form of 25(OH)D into its active form [29]. The importance of VDR in orthopedic problems was the subject of research by Muraki et al. (2011) [30]. The authors analyzed the association of serum vitamin D concentration and vitamin D receptor polymorphism (VDR) with knee pain and radiographic osteoarthritis (OA) of the knee in men and women in a large UK population cohort study. This study investigated the relationship of the Fok1, Cdx2, and Apa1 polymorphisms in the gene for VDR and serum 25(OH)D levels in 787 patients with a mean age of 65.6 ± 2.7 years. It has been proven that there was no association of Fok1, Cdx2, and Apa1 VDR polymorphisms with knee OA, except for Aa for Apa1 versus AA (*p* = 0.031). On the other hand, variants ff for Fok1 (*p* = 0.022) and AA for the Cdx2 polymorphism (*p* = 0.032) were significantly associated with a higher incidence of knee pain compared to FF for Fok1 and GG for Cdx2. The authors have not been able to reach an unequivocal conclusion. While it is known that vitamin D may be associated with pain severity, the evidence for an association between vitamin D genetic variability and knee OA pain is very weak in this study and should be expanded [30].

The role of the VDR gene polymorphism in pathophysiological processes is also emphasized by other scientists. Li et al. (2021) [31] performed a meta-analysis using allelic contrast, contrast of homozygotes, and recessive and dominant models to clarify the association between osteoarthritis (OA) and VDR ApaI, BsmI, TaqI, and FokI polymorphisms. Osteoarthritis (OA) is the most common joint disease affecting the knees, hips, hands, and spine, while its etiology remains unclear. Therefore, the authors considered 18 studies which included 2983 OA patients and 4177 controls. It was found that a statistically significant association exists between the VDR FokI polymorphism and OA susceptibility in the knee in the recessive model contrast (FF vs. Ff + ff; *p* = 0.028), while no significant associations were observed between the VDR ApaI polymorphism and OA. Ultimately, the authors concluded that the BsmI and TaqI VDR polymorphisms may be associated with a susceptibility to OA in the spine. However, the VDR ApaI polymorphism is not a significant genetic risk factor for osteoarthritis [31].

In turn, Hassan et al. (2022) [32] studied vitamin D receptor gene polymorphisms and the risk of osteoarthritis of the knee which may be correlated with TNF-α, a factor that inhibits macrophage migration (MIF), and 25-hydroxycholecalciferol concentration. The investigation included 205 participants which were assigned to the study (n = 105) and control group (n = 100). Additionally, the study group was divided into subgroups according to the severity of knee osteoarthritis (KOA) considered as mild, moderate, and severe, with 35 patients in each group. The authors applied the PCR-RFLP technique to SNP analysis for ApaI and TaqI, while vitamin D, serum and synovial TNF-α and MIF assays were performed using ELISA kits. It has been shown that KOA severity was correlated with significantly lower serum concentration of 25-hydroxycholecalciferol and significant increasing TNF-α and MIF levels (*p* ˂ 0.05). Authors noticed that wild homozygous and heterozygous mutant genotypes (GG + GT) and the G allele of ApaI demonstrated risk for KOA development. Additionally, the homozygous mutant CC genotype and C allele of TaqI may be considered as a risk factor in correlation with KOA development. As a result of the obtained results, it was concluded that VDR-SNPs, vitamin D3, TNF-α, and MIF have a potential function in the pathogenesis and progression of KOA with mechanistic associations [32]. Table 1 presents the most important research about the vitamin D metabolic pathway in total knee arthroplasty.

### 3.2. Hip Arthroplasty

Hip fractures are quite common in the elderly and most of these fractures are a consequence of osteoporosis and trauma [33]. Patients with a hip fracture are at risk of death. In case of injury, medical options include hemiarthroplasty (HA) which is a procedure of replacing the femoral head with a prosthesis, or total hip arthroplasty (THA) which includes femoral head and acetabulum replacement with prostheses [34]. Meta-analyses of studies that involved patients with a displaced hip fracture suggests that total hip arthroplasty results in less reoperation procedures than hemiarthroplasty [35,36,37]. However, not every study supports those clinical results, as there was no clinical difference with total hip arthroplasty or hemiarthroplasty in the group of 1495 randomly assigned patients [34]. The latest meta-analysis showed similar revision rate, function, mortality, periprosthetic fracture, and dislocation at up to 5 years [38]. In the analysis of Migliorini et al. (2022), total hip arthroplasty was compared to bipolar hemiarthroplasty (B-HHA) and unipolar hemiarthroplasty (U-HHA). THA scored similarly in terms of mortality but had lower rates of revision surgeries and lower rates of acetabular erosion, while mortality was positively associated with acetabular erosion [33].

Total hip arthroplasty (THA) is one of the most successful orthopedic surgeries and provides excellent pain relief and improvement of function [39]. Since its introduction in the 1960s, THA is still an excellent and reliable treatment procedure for the end stages of hip pathology. It gives satisfactory clinical outcomes up to 20 years follow-up [40].

The history of THA starts with metal-on-metal (MoM) bearings from 1955 to 1965. In the 1970s, metal-on-polyethylene (MoP) was introduced with a 77–81% success rate after primary THA. Additionally in 1970s, ceramic-on-ceramic (CoC) and ceramic-on-polyethylene (CoP) implants were used as competitive bearing alternatives along with MoM. The next step was using stainless steel, but only for short duration proposes. Nowadays, the main materials for THA are titanium, cobalt–chromium, PE, and ceramic. Furthermore, the most frequently used artificial hip joints that are used are composed of an acetabular cup, liner, head, and stem [41].

The development of materials made a huge contribution to the hip arthroplasty surgery procedures. For now, questions also arise: what influence do biochemical factors related to diet, supplementation, and metabolism have on the success of surgery, the recovery process, and the maintenance of physical condition after surgery?

#### 3.2.1. Vitamin D

The aim of the study of da Cunha et al. (2016) [42] was to investigate association of 25(OH)D among patients with total hip arthroplasty (THA). Among 158 screened patients, only 110 were included in this study. Excluded patients had at least one of the following conditions: autoimmune joint diseases, an active malignancy or systemic generic diseases, neurological motor diseases, and class 4 or 5 chronic kidney disease. Reasons for exclusion from this study were also: vitamin D dosing >50,000 IU per week, not undergoing postoperative evaluation after 3 months, and cases of hemiarthroplasty, revision arthroplasty, or arthrodesis conversion. Measurement of 25(OH)D2 and 25(OH)D3 were performed on peripheral venous blood samples using liquid chromatography with tandem mass spectrometry. The 25(OH)D level was found to have a weak correlation with changes in peak extension and peak power generation. The authors suggest that patients with higher 25(OH)D levels might have had a better muscle recovery after THA [42]. The role of vitamin D has been analyzed in vitro and in vivo and the results show vitamin D genomic and non-genomic effects. Further research was revolving around administration of vitamin D and acceleration of the functional restoration of the injured muscle as well as improving cellular turnover, cellular proliferation, and decreases in apoptosis after muscle injury. We suppose that not only vitamin D treatment as Stratos et al. suggested but also higher levels of vitamin D can increase P4 HB protein associated with collagen production [43]. Ongoing debate about serum concentrations of vitamin D and skeletal health prompted scientists to widen the knowledge about vitamin D levels and risk of hip arthroplasty for osteoarthritis (OA). A cohort study conducted by Hussain et al. [44], on 9135 patients showed a significant correlation between increasing serum 25(OH)D concentrations and an increased risk of hip arthroplasty for OA in males, when the same correlation did not show in the group of examined women. The authors declared that their results are independent inter alia of age, body mass index (BMI), smoking status, ethnicity, physical activity, and season of blood collection. The underlying mechanisms between serum 25OHD concentrations and the risk of hip OA requiring hip arthroplasty remain unclear, but it was suggested that increased bone mineral density (BMD) is also associated with vitamin D status and its regulatory effects in the bone forming process. Differences between males and females in incidence of hip OA are not clear [44]. Gender difference in serum vitamin D may be influential on other biochemical and hormonal metabolic pathways including sex steroids and associated with bone geometric structure change in the hip [45]. Another study of Visser et al. [46] included 87 patients that were scheduled for THA. A total of 23 patients were classified as vitamin D deficient, 32 patients as vitamin D insufficient, and 32 patients as vitamin D sufficient. The authors observed that vitamin D status before surgery was not associated with physical performance after surgery. It might be the result of sample collection seasons (September and March) when concentrations of vitamin D remain different [46]. Prevalence of low vitamin D status in patients undergoing THA was identified by Unnanuntana et al. [47], in a group of 200 patients comprised of 88 men and 112 women. Preoperative diagnoses were OA (187 patients), osteonecrosis (6 patients), hip dysplasia development (3 patients), and posttraumatic OA (4 patients). Almost 40% of the patients had low serum vitamin D levels. However, no correlation between serum vitamin D level and length of hospital stay or perioperative complications after THA were found. Later results of Unnanuntana et al., on the group of 219 enrolled patients showed that lower vitamin D status tended to occur among younger patients and those with higher BMI [47].

A study conducted by Maier et al. [48], found patients with vitamin D hypovitaminosis had a significantly longer length of stay compared to patients with normal serum vitamin D levels. A total number of 1083 patients were enrolled in this study, and they were scheduled to receive elective knee or elective hip arthroplasty. A total of 86% percent of the study population was vitamin D insufficient, and 63% were vitamin D deficient. Only 8% had a target range from 30 to 60 ng/mL. Serum 25OHD level was measured in the hospital laboratory. The results were that serum vitamin D and the length of stay in the orthopedic department were inversely related compared to patients with normal vitamin D levels in univariate analyses. In multivariate analyses, the length of stay remained significantly associated with 25OHD levels [48]. A study by Glowacki et al., on 68 women diagnosed with osteoarthritis of the hip showed no correlation between vitamin D level and BMD. However, twenty-two percent of the examined women had vitamin D deficiency, and three of them had an elevated parathyroid hormone level in serum. The study group was too small to draw conclusions [49]. Kim Hong Seok et al. (2021) compared the prevalence of vitamin D deficiency in elderly patients in two groups. The first group consisted of patients undergoing hip fracture surgery (HFS), and the second consisted of patients undergoing elective primary THA. This study included 70 patients (42 women, 28 men) in the HFS group, and 100 patients (74 women, 26 men) in the THA group. Serum 25OHD levels were measured by I125 radioimmunoassay. Serum vitamin D levels were less prevalent in patients undergoing THA than HFS. In the HFS group, vitamin D deficiency was more prevalent in sarcopenic than in non-sarcopenic patients. In 80% of HFS patients and 36% of THA patients, vitamin D insufficiency was detected. The authors referred to the studies showing the supplementation strategy of vitamin D which was effective and decreased the incidence of osteoporotic fractures in an elderly population [50,51]. The systematic review and meta-analysis of Yao et al., performed on 11 randomized clinical trials with 34,243 participants, showed that daily treatment with both vitamin D and calcium was a more promising strategy compared to standard dosing of vitamin D alone [52]. The metanalysis performed by Ghahfarrokhi et al. (2022) included 28 studies, 61,744 elderly people, and 9767 cases of hip fractures. Results of the analysis showed low serum vitamin D levels in the elderly were associated with an increase in the odds of hip fracture [53].

Traven et al. (2017) performed a retrospective review of 126 revision THA patients. Low vitamin D was not associated with a risk of 30-day readmission but was found to be associated with an increased risk of 90-day complications and periprosthetic joint infection as the reason for reversion surgery. The prevalence of vitamin D deficiency among the revision TJA population was significantly higher compared to the general population: 55% and 8%, respectively. Patients with low vitamin D levels experienced multiple postoperative complications more frequently (20.3%) than those with normal vitamin D levels (8.8%, *p* < 0.001) [54].

Nawabi et al., measured plasma 25(OH)D levels from patients with OA of the hip requiring total hip replacement (THR). Out of 62 patients, 15 were vitamin D insufficient/deficient (<40 nmol/L) and 47 patients were vitamin D sufficient (>40 nmol/L). The clinical state of patients was examined using the Harris hip score (HHS) before and after surgery. The vitamin D sufficient group had a significantly higher number of patients having excellent scores. To reduce the impact of any potential ceiling effect of the HHS, the difference in mean vitamin D levels were analyzed among patients scoring HHS 90 to 100 postoperatively with those who scored under 90 out of 100. Thirty-one patients had a postoperative HHS of >90 and the mean 25(OH)D level in this group was 67.5 nmol/L (SD 27.5). Thirty-one patients had a postoperative HHS of <90 with a mean 25(OH)D3 level of 48.6 nmol/L (SD 19.2) [55].

#### 3.2.2. Vitamin D Receptor (VDR)

The vitamin D receptor (VDR) has been linked in the biological cascade of events resulting in periprosthetic bone loss around loosened total hip replacements. A study suggested aseptic loosening and deep infection of the THR are the results of genetic influence of candidate susceptibility genes (VDR, IL-6, MMP1) and SNPs may be predictors of implant survival. About 150 patients in this case were genotyped for 3 genes. The TT genotype of VDR was statistically associated with osteolysis owing to deep infection as compared to the controls [56]. Studies describe the genotypes of VDR as a risk factor more than the level itself; however, the latest discovery of Topak et al. (2021) on the group of 80 children (40 healthy, 40 with dysplasia hip development (DDH)) showed significantly lower VDR levels among patients in the DDH group, while there was no difference in the levels of calcium (Ca), phosphorus (P), and alkaline phosphatase (ALP) observed [57].

Table 2 presents the characteristics of the vitamin D metabolic pathway components in hip arthroplasty.

### 3.3. Anterior Cruciate Ligament (ACL)

The anterior cruciate ligament (ACL) ensures proper stabilization of the knee and is a passive limiter of excessive translation of the anterior tibia. ACL injury causes pain, which significantly reduces the patient’s quality of life [58]. Despite many methods of treatment, surgical reconstruction of the ACL has remained the gold standard in the treatment of patients with ACL trauma for many years [59]. Unfortunately, according to estimates, every third patient will not return to the level of activity before the injury [63]. (Ardern et al., 2018). In order to increase the chances of returning to full fitness after ACL reconstruction, effective methods are sought to support the biological healing process of the transplant [63]. One of the major complications after ACL reconstruction is muscle weakness, which may persist for many years after the injury and may be a pediatric factor in osteoarthritis [60].

#### Vitamin D

Currently, more and more attention are paid to the role of vitamin D in the biological process of convalescence after ACL injury. Barker et al. (2011) [61] monitored the serum vitamin D levels of 18 men who underwent ACL surgical reconstruction. 25-hydroxyvitamin D status was assessed 2 weeks before and 3 months after surgery. Single-leg peak isometric forces were also performed. The control group consisted of 11 people without injuries. The researchers found that in 73% of those who did not have an ACL injury, the 25(OH)D concentration was less than 30 ng/mL. On the other hand, in the group of people with a damaged ACL, 56% had a 25(OH)D concentration less than 30 ng/mL 2 weeks before treatment, and 50% of patients in the months after the surgery. No statistically significant differences were found in the 25(OH)D content in the blood serum of beige and ACL trauma patients. Interestingly, peak isometric force was shown to be greater in the group of people after trauma and ACL reconstruction with baseline 25(OH)D levels higher than 30 ng/mL, compared to those with lower plasma vitamin D levels. The authors indicate that these results have great clinical value, as muscle weakness is a major rehabilitation challenge, and monitoring vitamin D levels can be a simple tool to alleviate strength deficits. After an ACL injury occurs, local inflammation is produced, accompanied by a cytokine release. Interferon γ (IFNγ) is a pro and anti-inflammatory cytokine and it mediates the enzymatic conversion of 25(OH)D to the hormonal form of vitamin D (1,25 (OD) D) by altering the activity of 1α-hydroxylase [62]. Interesting data on the correlation of IFNγ with vitamin D concentration was provided by a study on a group of 12 men after ACL injury [61]. The researchers collected plasma from patients before ACL reconstruction surgery and 90 min, 3 days, and 7 days after surgery. It was shown that immediately after the procedure the concentration of 25(OH)D decreased significantly, which was accompanied by a significant increase in IFNγ activity. In the following days, the level of these parameters returned to normal. The correlation between IFNγ and vitamin D metabolites was also examined, and it was found that despite the inverse correlation of the 25(OH)D change, changes in IFNγ were correlated with the concentration of 1,25(OH)D and the ratio of 1,25(OH)D to 25(OH)D both before and after the ACL operation. These data suggest that in the future, in addition to 25(OH)D, the 1,25(OH)D form should also be determined in patients. Further evidence for the role of vitamin D in muscle regeneration was provided by the studies by Barker et al., 2013 [64]. Researchers analyzed the levels of 25(OH)D and other biochemical parameters in the serum of 14 healthy people after intense leg training. The mean concentration of 25(OH)D in the study group was 28.0 ± 2.5 ng/mL. In 36% of the subjects, the serum 25(OH)D concentration was higher than 32 ng/mL, and 64% had a concentration below this level. It was found that eccentric muscle shortening caused a significant increase in serum 25(OH)D levels immediately after training, and this level was leveling off just minutes after exercise. Peak isometric strength was also observed; it significantly decreased immediately after training and was maintained for several days. The biochemical analysis showed that the content of IFNγ, calcium, and serum albumin also increased immediately after the training. The authors suggest that the determination of serum 25-hydroxyvitamin D concentration may be a determinant of muscle strength after training or trauma.

Gupta et al., 2021 monitored vitamin D levels in people after ACL reconstruction. The study included 153 people who were under the supervision of a doctor for two years after the procedure. The patients were divided into three groups depending on the content of 25(OH)D (first group—<20 ng/mL; second group—20–30 ng/mL; third group—<30 ng/mL). The authors showed that there was no correlation between serum vitamin D levels and the effectiveness of graft acceptance during ACL reconstruction. The graft acceptance failure rates in the study groups were 5.88%, 1.96%, and 1.96% (first, second, and third groups, respectively). Unfortunately, the authors did not describe in their work how rehabilitation proceeded after the surgery [65].

Interesting observations are provided by the experiment carried out on mice by Wolf et al., 2019 [66]. They used mice to create a model of post-traumatic osteoarthritis. A small group of animals underwent ACL reconstruction to assess the degree of degenerative disease induction. All animals received four levels of dietary vitamin D supplementation. The authors observed that with the increase in the dose of vitamin D in the diet, the content of 25(OH)D in the plasma increased and the amount of vitamin D binding protein (DBP) in the plasma decreased. In this study, no correlation was found between the vitamin D dose and osteoarthritis. On the other hand, the protective effect of vitamin D was demonstrated. Mice developed osteoarthritis, however, in females who received superphysiological doses of vitamin D, it was milder [66]. Table 3 presents the characteristics of the vitamin D metabolic pathway components in anterior cruciate ligament arthroplasty.

### 3.4. Rotator Cuff (RC)

Rotator cuff (RC) injuries are the most common cause of shoulder pain and dysfunction [68]. In general, the recommended method of treating this disease is surgery. However, the failure rate of this procedure is as high as 94% [69]. After a rupture, the muscle recedes less than the tendons. Local inflammation develops in the muscle cells at the site of injury. Cells release profibrotic factors, which are key regulators of gene expression. This early inflammatory response leads to tenocyte apoptosis and degradation of muscle fibers. This leads to the regeneration of the muscle [70]. At the tendon–bone interface, macrophages cause the formation of scar tissue, but not normal tendon tissue, by secreting myogenic regulatory factors [71]. The complex mechanism of interaction at the molecular and cellular level leads to further scar tissue formation and permanent irreversible changes in the RC structure [69].

#### Vitamin D

Lee et al. (2021) [72] conducted a retrospective study on a large group of people who had undergone rotator cuff repair. The concentration of 25(OH)D was measured in patients at the time of qualification for surgical treatment of the injury. The researchers assumed that hypovitaminosis was a serum 25(OH)D concentration below 20 ng/mL. The prevalence of hypovitaminosis D in patients with rotator cuff tears was found to be 44.3%, and 29% of patients had normal vitamin D levels (>30 ng/mL). The mean serum concentration of 25(OH)D among all patients was 24.7 ± 13.7 ng/mL. Additionally, it was found that serum 25(OH)D concentration was positively correlated with age [72].

25(OH)D directly influences intracellular phosphate accumulation by muscle cells and plays a significant role in maintaining muscle metabolism and function [73]. Fat degeneration is a frequently observed lesion after rotator cuff detachment [74]. Oh et al., 2009 assessed the correlation between the fatty degeneration of the muscle and the level of vitamin D in the serum of people with shoulder joint disorders. The study included 228 patients with a full-thickness tear (group 1) and 138 patients with other conditions of the shoulder (group 2). Serum 25(OH)D concentration was measured in all participants and the fatty degeneration of supraspinatus, infraspinatus, and subscapularis was measured using magnetic resonance arthrography. The mean serum 25(OH)D level was shown to be 44.02 ng/mL and 43.64 ng/mL (group 1 and group 2, respectively). In both groups, about half of the respondents had vitamin D hypovitaminosis (<20 ng/mL). The analysis of the collected data allowed the authors to conclude that lower levels of vitamin D in the serum was associated with higher fatty degeneration in the muscles of the cuff. Spearman’s correlation coefficients were 0.173 (*p* = 0.001), −0.181 (*p* = 0.001), and −0.117 (*p* = 0.026) for supraspinatus, infraspinatus, and subscapularis, respectively [73]. There are studies that emphasize that muscle fat hydration is an irreversible prognostic indicator [74]. Therefore, more research is needed to determine if and how vitamin D levels will affect the degree of fatty degeneration in muscle in the RC [73].

Rhee et al. in 2020 [75] examined the correlation between vitamin D concentration on the day of surgery, vitamin D receptor, and postoperative vitamin D concentration with muscle performance, fatty degeneration, and healing failure one year after RC reconstruction surgery. A total of 26 patients participated in the study. The mean serum 25(OH)D level one year after the procedure was 20.5 ± 9.2 ng/mL. Only 23.1% of patients had normal vitamin D levels (>20 ng/mL). It was found that patients who had lower serum vitamin D levels before surgery also had lower serum levels one year after surgery. People with higher levels of vitamin D preoperatively and one year after surgery also had a lower index of the isokinetic muscle efficiency test. Additionally, no correlation was found between vitamin D concentration and muscle fat degeneration, and between the vitamin receptor and other parameters studied. These studies indicate that vitamin D supplementation in patients with low levels of this vitamin before surgery could have a positive effect on rotator cuff muscle performance after surgery [75].

Another team that looked for a link between vitamin D levels and muscle fat degeneration was Ryu et al., 2015 [76]. The study involved 91 patients who underwent arthroscopic rotator cuff repair for full-thickness, small-sized to massive tears were evaluated. Serum 25(OH)D levels were assessed in all patients, and the structure and function of the muscle was assessed after surgery. Preoperative 25(OH)D concentration was normal in only 3% of patients (>30 ng/mL), and as many as 88% of patients had vitamin D hypovitaminosis (<20 ng/mL). A good source for supplementing the current state of knowledge are pilot studies on animal disease models [76]. Angeline et al., 2014 [77] analyzed the effect of vitamin D deficiency on the structure of the healing tendon–bone interface in rats. Only male rats induced with vitamin D deficiency were included in the study. Then, after 6 weeks, they underwent a unilateral right tendon detachment of the supraspinatus followed by immediate surgical repair of the injury. The effects were assessed 2 and 4 weeks after the treatment. The conducted experiment showed no correlation between low vitamin D concentration and total mineral density and fraction of cortical bone volume, whole, or spongy bone 4 weeks after surgery. Biomechanical testing demonstrated a significant decrease in load to failure in the experimental group compared with controls at 2 weeks (5.8 6 2.0 N vs. 10.5 6 4.4 N, respectively; *p* = 0.006). Histological analysis showed less bone formation and less collagen fiber organization in the vitamin D deficient specimens at 4 weeks as compared with control. The obtained data suggest that vitamin D hypovitaminosis may adversely affect healing at the rotator cuff repair site. The authors emphasize that more research is needed to identify the mechanism by which vitamin D influences tendon healing and whether vitamin D supplementation can effectively influence rotator cuff healing and reduce the incidence of relapses [77]. Table 4 presents the characteristics of the vitamin D metabolic pathway components in rotator cuff arthroplasty.

### 3.5. Shoulder Arthroplasty

#### Vitamin D

The importance of vitamin D deficiency is also emphasized in patients undergoing total shoulder arthroplasty. Smith et al. (2021) [78] conducted a study on the correlation of vitamin D deficiency with complications after shoulder arthroplasty. Using the PearlDiver database, 1674 patients with vitamin D deficiency were analyzed and compared with 5022 controls. The authors considered complications related to the implant, such as loosening, periprosthetic fracture, periprosthetic joint infection, and revision shoulder arthroplasty. There was a significantly higher rate of revision shoulder arthroplasty in patients with vitamin D deficiency compared to the control group (2.3% vs. 0.8%, odds ratio 3.3, *p* < 0.0001); however, no significant differences were found in any of the other complications [78].

Analyses of vitamin D hypovitaminosis in patients undergoing shoulder arthroplasty were also undertaken by Inkrott et al. (2016) [79]. The study included a total of 218 people who underwent one of the operations: (I) total shoulder arthroplasty (TSA), (II) inverse total shoulder arthroplasty (RSA), and (III) hemi shoulder arthroplasty (HA). The authors noted that the vast majority of shoulder arthroplasty patients had vitamin D deficiencies at least < 30 ng/mL, and the phenomenon was increased in patients with a high body mass index (BMI of 30 kg/m^2^ or greater). According to the American Society of Anesthesiologists (ASA), age, gender, race, and smoking were irrelevant, and a lack of supplementation with vitamin D and calcium was the most significant risk factor for hypovitaminosis D [79]. Table 5 presents the characteristics of the vitamin D metabolic pathway components in shoulder arthroplasty.

We are aware of many limitations in our study. The first limitation is that there are inconsistent articles that do not include the seasonality of vitamin D and its levels in humans making it impossible to compare results. Only a few articles stated the season of biological material collection. We suggest that it should be included in every study that concentrates on components of the vitamin D metabolic pathway. There were also a few studies that did not include important data about the method of identification/analysis or biological material (blood/plasma/serum and fasting/non-fasting). We showed a lack of information in the summary tables. Taking into consideration the short half-life of vitamin D metabolites, we suggest including more information about the time of measurement after receiving a sample. The third limitation was using only 2010–2022 articles to possess the newest data; however, in the last decade, prostheses materials have improved, and we took into consideration the fact that it also might be one of the factors in better healing in many orthopedic procedures and treatments.

## 4. Conclusions

Vitamin D and its metabolic pathway is an important aspect in orthopedics. The processes during the healing of fractures, stabilization of joint prostheses, and the healing of soft tissues requires two equally crucial factors, which are mechanical stabilization and biological processes. The current knowledge on the role of anastomosis stability and arthroplasty is well known and is still developing; however, research on the biology of repair processes is limited to finding the important role of vitamin D in the metabolism of bone tissue. In our opinion, vitamin D deficiency should be considered comprehensively. The vast majority of research articles concern the metabolite of vitamin D (25(OH)D), which is usually measured in diagnostic laboratories. However, it is also worth considering other important components of the pathway including vitamin D binding protein (VDBP), cytochromes (CYP24A1/CYP27B1/CYP2R1), and the vitamin D receptor (VDR) as biochemical (concentration) and as genetic (polymorphisms in genes, genes expression) parameters.

Despite the fact that there is much valuable information in the literature, we believe that the other elements of the vitamin D pathway also deserve attention, and their role in correlation with orthopedic disorders should be assessed to supplement the missing knowledge on this topic.

## Figures and Tables

**Figure 1 ijms-23-15556-f001:**
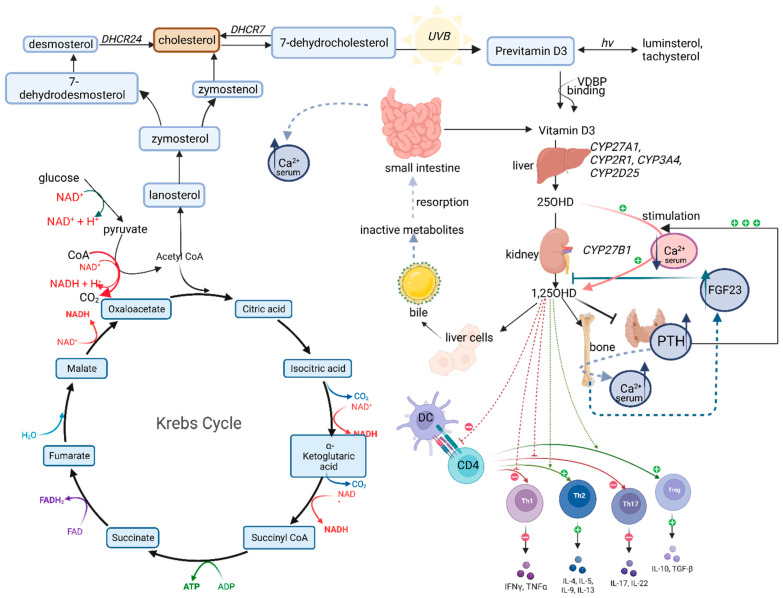
Metabolic pathway of vitamin D (created with BioRender.com, accessed on 18 September 2022).

**Figure 2 ijms-23-15556-f002:**
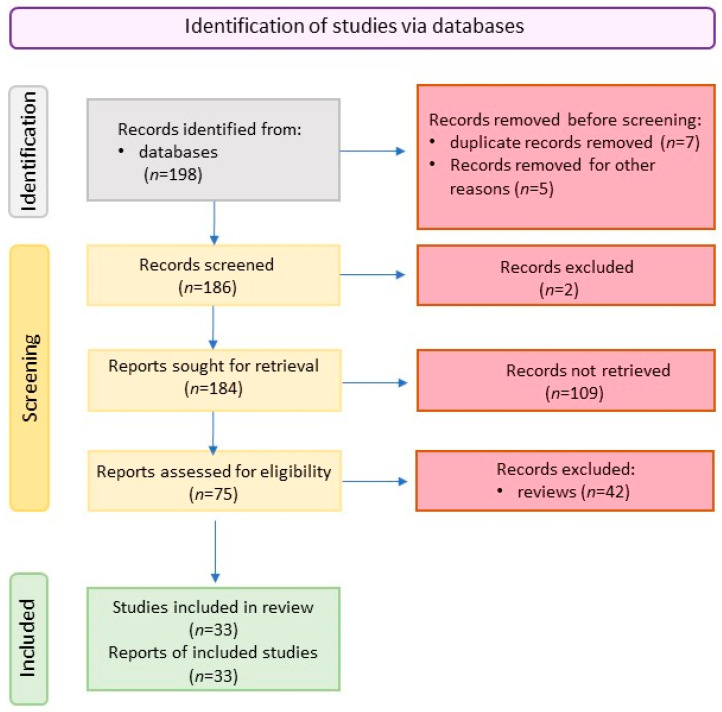
Scheme of literature searching.

**Table 1 ijms-23-15556-t001:** Characteristic of vitamin D components in total knee arthroplasty.

Component of Pathway	Number of Patients	Biological Material	Method of Identification/Analysis	Results	References
Vitamin D	n = 92	Serum	I^125^ radioimmunoassay	Patients with normal preoperative vitamin D levels showed better results of physical activity; therefore, it seems that vitamin D deficiency adversely affects early postoperative functional results after TKA	[17]
Vitamin D	n = 120	Serum	No data	TKA should be not delayed among patients with vitamin D deficiency. Use of appropriate postoperative supplementation is able to improve recovery in patients	[18]
Vitamin D	n = 174	Serum	No data (analysis done as a service in the diagnostic laboratory)	Supplementation of vitamin D should be used in patients prior to TKA. A loading dose regimen of 50,000 IU weekly for 4 weeks followed by a maintenance dose of 2000 IU/d more effectively improves vitamin D deficiency in comparison to a low-dose, daily regimen among TKA patients	[19]
Vitamin D (+ calcium)	n = 142,147	Serum	No data (analysis done as a service in the diagnostic laboratory)	After TKA, implant survival was significantly better in people taking a combination of calcium and vitamin D (at a dose of 800 IU or more) for more than 1 year compared to those who had never used it	[20]
Vitamin D	n = 226 (postmenopausal women)	Serum	No data (analysis done as a service in the diagnostic laboratory)	Vitamin D deficiency may adversely affect early functional outcomes in postmenopausal women after TKA and may be considered as a risk factor for moderate to severe postoperative knee pain (in correlation with smoking and high mass index)	[21]
Cholesterol (+ apolipoprotein A1)	n = 20	Serum	No data (analysis done as a service in the diagnostic laboratory)	Vitamin D deficiency is associated with a higher rate of all-cause revision TSA but not medical complications compared to controls	[22]
Total triglycerides (TG), total cholesterol (TC), high-density lipoprotein cholesterol (HDL-C), low-density lipoprotein cholesterol (LDL-C), apolipoprotein A1 (ApoA1), and apolipoprotein B (ApoB)	n = 184	Serum, synovial fluid	No data (analysis done as a service in the diagnostic laboratory)	Any significant differences were not determined in serum TG and ApoB concentrations between both groups, while primary knee osteoarthritis patients had higher TC and LDL-C levels, and lower HDL-C and ApoA1 levels	[23]
Vitamin D binding protein (VDBP) (+ 25(OH)D, parathyroid hormone (PTH), calcium, C-reactive protein (CRP), and albumin)	n = 33	Serum	No data (analysis done as a service in the diagnostic laboratory)	A molar ratio of vitamin D to VDBP should be considered to provide a useful indicator of biological activity in TKA aspect	[24]
Vitamin D metabolites and sex steroids (included total and free serum concentrations of 25OHD, 1,25(OH)2D3, 24,25-dihydroxyvitamin-D, DBP, albumin, sex hormone binding globulin (SHBG), calcium, and parathyroid hormone (PTH))	n = 25	Serum	No data (analysis done as a service in the diagnostic laboratory)	The concentration of serum albumin and SHBG decreased postoperatively. Unexpectedly, the concentrations of DBP and 25OHD remained unchanged, but 1,25(OH)2D3 decreased after surgery. 1,25(OH)2D3 was lower (about 24%) 3 weeks postoperatively compared to preoperative levels, while 24,25-dihydroxyvitamin-D was unchanged in postmenopausal women. The calculated conversion ratio of 25OHD to 1,25(OH)2D3 was strongly related to the preoperative effects of serum 25-OHD and PTH, while serum calcium was the most predictive after surgery	[25]
CYP27B1 and CYP24A1 activity in vitamin D pathway	n = 25	Serum	No data on the method of determination (analysis done as a service in the diagnostic laboratory)	High activity of CYP24A1 in women concomitantly with lower CYP27B1 levels means that a greater proportion of the substrate pool is inactivated due to increased CYP24A1 activity, which alone or in combination with decreased CYP27B1 activity may result in the low 1,25(OH)2D3	[25]
Relationship of the Fok1, Cdx2, and Apa1 polymorphisms in the gene for VDR and serum 25(OH)D levels	n = 787	Serum/blood	Chemiluminescent assay with equal specificity for both D2 and D3; VDR genotype was determined by the polymerase chain reaction (PCR) and restriction fragment length polymorphism (RFLP) analysis (polymorphic sites for Fok1, Cdx2, and Apa1)	Vitamin D may be associated with pain severity, the evidence for an association between vitamin D genetic variability and knee OA pain is very weak in this study and should be expanded	[30] *
Vitamin D receptor gene polymorphisms (+level of TNF-α, a factor that inhibits macrophage migration (MIF), and 25-hydroxycholecalciferol)	n = 205	Serum/blood	PCR-RFLP was applied to SNP analysis for ApaI and TaqI, while vitamin D, serum and synovial TNF-α and MIF assays were performed using ELISA kits	KOA severity was correlated with significantly lower serum concentration of 25-hydroxycholecalciferol and significant increasing TNF-α and MIF levels	[32] **

* Knee pain and radiographic osteoarthritis (OA) of the knee; ** Knee osteoarthritis (KOA).

**Table 2 ijms-23-15556-t002:** Characteristics of the vitamin D metabolic pathway components in hip arthroplasty.

Component of Pathway	Number of Patients	Biological Material	Method of Identification/Analysis	Results	References
Vitamin D	158, included 110	No data	Liquid chromatography with tandem mass spectrometry detection	(1) 25(OH)D levels were correlated with change in peak extension and peak power generation. (2) The effect of 25(OH)D on the change in these variables is modest. Studies with longer follow-up are warranted to establish the role of vitamin D in THA rehabilitation	[58]
Vitamin D	11,247 people (national, population-based cohort study)	Fasting serum or fasting plasma	Competitive chemiluminescent immunoassay with an interassay coefficient of variation	Increasing serum 25-hydroxy-vitamin D concentrations were associated with an increased risk of hip arthroplasty for OA in males, while no significant association was observed in females	[59]
Vitamin D	n = 87	No data	High-performance liquid chromatography	Vitamin D status did not appear to affect physical recovery after THA. The drop in vitamin D after surgery deserves further investigation, but could possibly be explained by hemodilution	[46]
Vitamin D	n = 200 (88 men, 122 women)	Serum	No data	(1) From 200 patients, 79 (39.5%) had low serum vitamin D (serum 25-hydroxy vitamin D < 32 ng/mL). (2) There were no associations between serum vitamin D level and the attainment of in-hospital functional milestones as well as length of hospital stay or perioperative complications after THA	[39]
Vitamin D	n = 1083 (567 women, 516 men)	Serum	ARCHITECT^®^ 25-OH vitamin D assay	(1) A total of 86% of the study population was vitamin D insufficient and 63% of patients were vitamin D deficient. Of the 1083 vitamin D serum levels measured in this study, only 8% were in the target range of 30 to 60 ng/mL. Serum vitamin D levels of all 1083 patients were normally distributed, with a mean of 17.1 ng/mL. (2) The length of stay was longer in patients with hypovitaminosis D compared to patients with normal serum 25-OH-D levels	[48]
Vitamin D	n = 219	Non-fasting serum samples from blood	Radioreceptor assay	(1) Of 219 patients, 102 (46.6%) had low vitamin D levels (25-hydroxyvitamin D < 30 ng/mL). Low vitamin D status did not adversely affect short-term function at 6 weeks after THA. (2) There was no association between serum vitamin D levels and the within-patient changes of scores of each outcome measurement	[47]
Vitamin D	n = 100	Serum	I125 radioimmunoassay	The level of serum vitamin D was lower, and the percentages of vitamin D insufficiency and deficiency patients were higher in the HFS patients compared to those in the THA patients	[60]
VDBP status	n = 64	Peripheral venous blood sampling	No data	(1) Acute PPIs of the hip and knee joints show a significantly reduced calcium and 25 OH vitamin D3 levels as well as lowered proteins (albumin and total protein) compared with chronic infections as well as primary endoprostheses and aseptic replacement operations. (2) Substitution of vitamin D3 and calcium with simultaneous adaptation of the protein balance is recommended in all PPIs, especially in the acute PPI	[61]
VDBP status	n = 103 (65 females, 38 males,	Bone mRNA levels	Bone mRNA levels for vitamin D metabolizing enzymes CYP27B1 and CYP24A1 were measured by qRT-PCR.	(1) Serum 25(OH)D levels were associated with MWT with values significantly greater in patients with higher serum 25(OH)D levels. (2) Serum 25(OH)D levels were negatively associated with bone surface/bone volume (BS/BV), and together with bone CYP27B1 and CYP24A1 mRNA accounted for 10% of the variability of BS/BV	[62]
CYP24A1					

**Table 3 ijms-23-15556-t003:** Characteristics of the vitamin D metabolic pathway components in anterior cruciate ligament arthroplasty.

Component of Pathway	Number of Patients	Biological Material	Method of Identification/Analysis	Results	References
25-hydroxyvitamin D	CTR *n* = 11(non-injured); ACL *n* = 18 (injured)	Plasma	Chemiluminescent assay	(1) In 73% of CRT, the content of 25-hydroxyvitamin D was lower than 30 ng/mL. (2) 2 weeks before surgery, 56% ACL had content of 25-hydroxyvitamin D lower than 30 mg/mL, and 50% ACL 3 min after surgery	[67]
25-hydroxyvitamin D; 1.25-hydroxyvitan D;	*n* = 12	Plasma	Chemiluminescent assay	(1) 90 min after the injury, the concentration of 25(OH)D decreased significantly and IFNγ activity significant increased. (2) Despite the inverse correlation of the 25(OH)D change, changes in IFNγ were correlated with the concentration of 1.2 (OH D and the ratio of 1,25(OH)D to 25(OH)D both before and after ACL reconstruction	[62]
25-hydroxyvitamin D	*n* = 14, after intense exercise	Plasma	Chemiluminescent assay	(1) Mean concentration of 25(OH)D was 28.0 ± 2.5 ng/mL; in 36% of patients, the 25(OH)D concentration was higher than 32 ng/mL, and 64% were below this level. (2) Intense exercise caused a significant increase in serum 25(OH)D levels immediately after training	[64] *
25-hydroxyvitamin D	*n* = 153 (group 1 25(OH)D <20 ng/mL n = 51; group 2 25(OH)D 20–30 ng/mL n = 51; group 3 25(OH)D >30 ng/mL n = 51)	Serum	Chemiluminescent assay	Graft acceptance failure rates in the study groups 1, 2, and 3 were 5.88%, 1.96%, and 1.96%, respectively. There was no correlation between serum vitamin D levels and the effectiveness of graft acceptance after ACL reconstruction	[65]
25-hydroxyvitamin D; D-binding protein (DBV)	C57-BL6 mice	Plasma	ELISA	(1) The increase in the dose of vitamin D in the diet resulted in a higher content of 25(OH)D in the plasma and lower amount of vitamin D-binding protein (DBP). (2) There was no correlation between the vitamin D dose and osteoarthritis, but the protective effect of vitamin D was demonstrated. In females who received superphysiological doses of vitamin D, it was milder	[66]

* No diagnosis of the disease, the study group are healthy people after intense training.

**Table 4 ijms-23-15556-t004:** Characteristics of the vitamin D metabolic pathway components in rotator cuff arthroplasty.

Component of Pathway	Number of Patients	Biological Material	Method of Identification/Analysis	Results	References
25-hydroxyvitamin D	n = 176	Serum	No data	(1) A total of 44% of patients had hypovitaminosis of vitamin D (content of 25(OH)D < 20 ng/mL) and 29% had normal levels of vitamin D (content of 25(OH)D > 30 ng/mL). (2) Mean serum concentration of 25(OH)D among all patients was 24.7 ± 13.7 ng/mL. (3) 25(OH)D concentration was positively correlated with age	[72]
25-hydroxyvitamin D; fatty degeneration	Group 1: patients with a full-thickness tear n = 228; Group 2: patients with other conditions of the shoulder n = 138	Serum	25(OH)D was measured using the radioimmunoassay test. The fatty degeneration of supraspinatus, infraspinatus, and subscapularis was measured with themagnetic resonance arthrography	(1) Mean serum 25(OH)D level was shown to be 44.02 ng/mL (group 1) and 43.64 ng/mL (group 2). (2) A lower level of vitamin D in the serum was associated with higher fatty degeneration in the muscles of the cuff	[73]
25-hydroxyvitamin D; vitamin D receptor (VDR)	n = 26	Serum	25(OH)D was assessed using liquid chromatography; VDR was measured by western blotting	(1) Mean serum 25(OH)D level one year after injured was 20.5 ± 9.2 ng/mL. A total of 23.1% of patients had normal vitamin D levels (>20 ng/mL). (2) Patients who had lower 25(OH)D levels before surgery also had lower serum levels one year after surgery. (3) Patients with higher levels of vitamin D preoperatively and one year after surgery had a lower index of the isokinetic muscle efficiency test. (4) There was no correlation between vitamin D concentration and muscle fat degeneration, and between the vitamin receptor VDR and other parameters studied	[75]
25-hydroxyvitamin D	n = 28 (male Sprague–Dawley rats)	Serum	No data	(1) There was no correlation between low vitamin D concentration and total mineral density and fraction of cortical bone volume, whole, or spongy bone 4 weeks after surgery. (2) Histological analysis showed less bone formation and less collagen fiber organization in the vitamin D deficient specimens at 4 weeks as compared with control. (3) The data suggest that vitamin D hypovitaminosis may adversely affect healing at the rotator cuff repair site	[77]

**Table 5 ijms-23-15556-t005:** Characteristics of the vitamin D metabolic pathway components in shoulder arthroplasty.

Component of Pathway	Number of Patients	Biological Material	Method of Identification/Analysis	Results	References
Vitamin D	n = 5022	Serum	No data on the method of determination (analysis done as a service in the diagnostic laboratory)	(1) There was a significantly higher rate of revision shoulder arthroplasty in patients with vitamin D deficiency compared to the control group. (2) No significant differences were found in any of the other complications	[78]
Vitamin D	n = 218	Serum	No data on the method of determination (analysis done as a service in the diagnostic laboratory)	The authors noted that the vast majority of shoulder arthroplasty patients had vitamin D deficiencies at least <30 ng/mL, and it was increased in patients with a high body mass index	[79]

## Data Availability

Not applicable.

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
