# Peer review of "Vitamin D Metabolic Pathway Components in Orthopedic Patientes—Systematic Review"

_ijms, 2022, doi:10.3390/ijms232415556_

Round 1
Reviewer 1 Report
This study could have included series in which there was active Vitamin D supplementation as part of the study protocol. It would be interesting to see the difference in outcomes. The inclusion of Vitamin D supplementation studies is quite important, in my view.
Author Response
The authors would like to thank to Reviewer for taking the time reviewing this work.
Reviewer 2 Report
In the manuscript, the authors conducted a review for the role of Vitamin D metabolic pathway components for the bone related diseases. The review is comprehensive, and illustrate the metabolite of vitamin D (25 (OH) 738 D) in the published literatures. Here are some suggestions and comments:
(1) The fracture healing is also accompanied by the formation of vessel. Did the author find any relationship between Vitamin D and vessel?
(2) In the conclusion, the authors point out that other elements of the vitamin D pathway are required to be investigate, could the author list them in details?
(3) Could the author further clarify the future research topic in the area of Vitamin D and bone disease?
Author Response
1. Question: The fracture healing is also accompanied by the formation of vessel. Did the author find any relationship between Vitamin D and vessel?
Answer: Vitamin D is linked to many processes in the body. In this paper, we focused on its relationship to orthopedics with particular emphasis on the components of the vitamin D pathway.
The vitamin D topic is very broad, so in this manuscript we have focused on these aspects in order to keep our view as clear as possible. Thank you very much for the hint related to the process of fracture healing - we will look at this topic and include it in the next publication. We are planning a series of publications related to the role of vitamin D and we will present this topic more detailed.
2. Question: In the conclusion, the authors point out that other elements of the vitamin D pathway are required to be investigate, could the author list them in details?
Answer: Thank you for this suggestion. We added the information in the manuscript (lines 738-743).
In our opinion, vitamin D deficiency should be considered comprehensively. It is also worth considering other important components of the pathway: vitamin D binding protein (VDBP), cytochromes (CYP24A1/CYP 27B1/CYP2R1), Vitamin D receptor (VDR) – as biochemical (concentration), and as genetic (polymorphisms in genes, expression) parameters.
3. Question: Could the author further clarify the future research topic in the area of Vitamin D and bone disease?
Answer: We plan to undertake comprehensive research on the role of the vitamin D pathway in selected orthopedic diseases. Our research will focus on the components of the pathway: vitamin D binding protein (VDBP), Cytochromes (CYP24A1/CYP 27B1), Vitamin D receptor(VDR), cholesterol. We plan, among others, to perform experiments in vitro using cell lines and clinical material taken from patients, also biochemical, and genetic correlations, however, we would like to include a detailed description and goals in the next publication.
Round 2
Reviewer 2 Report
Thank you for the revision. It addressed my concerns.